# E2E-MFD: Towards End-to-End Synchronous Multimodal Fusion Detection

**Jiaqing Zhang**[1]**, Mingxiang Cao**[1]**, Weiying Xie**[1*]**, Jie Lei**[2]**,**
**Daixun Li**[1]**, Wenbo Huang**[3]**, Yunsong Li**[1]**, Xue Yang**[4*]

[1]The State Key Laboratory of Integrated Services Networks, Xidian University
[2]University of Technology Sydney    [3]Southeast University    [4]Shanghai AI Laboratory

https://github.com/icey-zhang/E2E-MFD

## Abstract

Multimodal image fusion and object detection are crucial for autonomous driving. While current methods have advanced the fusion of texture details and semantic information, their complex training processes hinder broader applications. Addressing this challenge, we introduce E2E-MFD, a novel end-to-end algorithm for multimodal fusion detection. E2E-MFD streamlines the process, achieving high performance with a single training phase. It employs synchronous joint optimization across components to avoid suboptimal solutions associated to individual tasks. Furthermore, it implements a comprehensive optimization strategy in the gradient matrix for shared parameters, ensuring convergence to an optimal fusion detection configuration. Our extensive testing on multiple public datasets reveals E2E-MFD's superior capabilities, showcasing not only visually appealing image fusion but also impressive detection outcomes, such as a 3.9% and 2.0% mAP$_{50}$ increase on horizontal object detection dataset M3FD and oriented object detection dataset DroneVehicle, respectively, compared to state-of-the-art approaches.

## 1 Introduction

Precise and reliable object parsing is critical in fields such as autonomous driving [1] and remote sensing monitoring [2]. Relying solely on visible sensors can lead to inaccuracies in object recognition in challenging environments, like inclement weather conditions. Visible-infrared image fusion [3; 4; 5; 6] as a typical common multimodal fusion (MF) task addresses these challenges by leveraging complementary information from different modalities, leading to the rapid development of various multimodal image fusion techniques [7; 8; 9; 10; 11]. Techniques like CDDFuse [12] and DIDFuse [13] employ a two-step process where a MF network is trained initially, followed by training an object detection (OD) network with the results from the MF network to assess fusion effectiveness separately. Although deep neural networks have significantly enhanced the ability to learn representations across modalities, resulting in promising multimodal fusion outcomes, the focus has predominantly been on producing visually appealing images. This emphasis often overlooks the improvement of downstream high-level visual tasks, such as enhanced object parsing, which remains a substantial hurdle.

Recent studies have devoted into designing joint learning methods that integrate fusion networks with high-level tasks such as object detection [14] and segmentation [15; 16]. The synergy between MF and OD in Multimodal Fusion Detection (MFD) methods has emerged as a vibrant area of research. This partnership allows MF to produce richer, more informative images, enhancing OD performance, while OD contributes valuable object semantic insights to MF, aiming to accurately locate and identify objects in a scene. Typically, MFD networks adopt a cascaded design where joint

---

[*]Corresponding author

38th Conference on Neural Information Processing Systems (NeurIPS 2024).

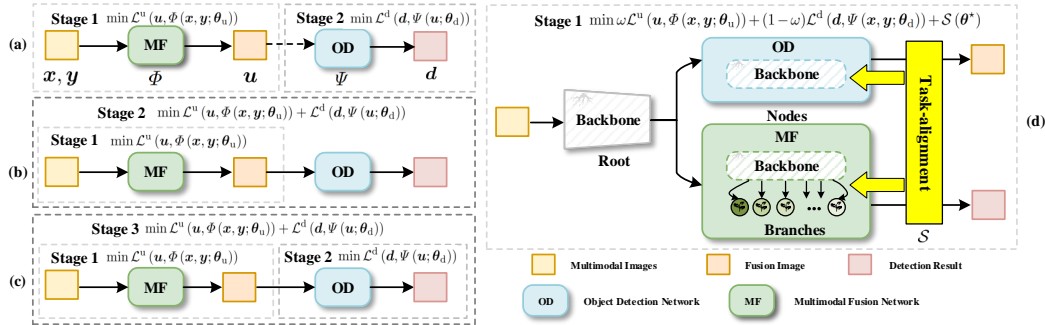

Figure 1: Comparison of (d) E2E-MFD with existing MF-OD task paradigms (a) Two-Stage (Separate Cascaded), (b) Two-stage (Joint Cascaded) and (c) Multi-stage (Joint Cascaded).

optimization techniques [17] use the OD network to guide the MF network toward creating images that facilitate easier object detection. Notably, Zhao *et. al* [18] introduced a joint learning method for multimodal fusion detection, incorporating meta-feature embedding from OD to improve fusion by generating semantic object features through meta-learning simulation. Despite these advancements, as highlighted in Figure 1, significant challenges persist: 1) Current optimization approaches rely on a multi-step, progressively joint method, compromising efficiency; 2) These methods overly focus on leveraging OD information for fusion enhancement, leading to difficulty in parameter balancing and susceptibility to local optima of individual tasks. Therefore, the quest for a unified feature set that simultaneously caters to each task remains formidable.

In this paper, we introduce E2E-MFD, an end-to-end algorithm for multimodal fusion detection, designed to seamlessly blend detailed image fusion and object detection from coarse to fine levels. E2E-MFD facilitates the interaction of intrinsic features from both domains through synchronous joint optimization, allowing for a streamlined, one-stage process. To reconcile fine-grained details with semantic information, we propose the novel concept of an Object-Region-Pixel Phylogenetic Tree (ORPPT) coupled with a coarse-to-fine diffusion processing (CFDP) mechanism. This approach is inspired by the natural process of visual perception, tailored to meet the specific needs of MF and OD. Furthermore, we introduce a Gradient Matrix Task-Alignment (GMTA) technique to fine-tune the optimization of shared components, thereby minimizing the adverse impacts traditionally associated with inherent optimization challenges. This ensures an efficient convergence towards an optimal set of fusion detection weights, enhancing both the accuracy and efficacy of multimodal fusion detection.

Our contributions in this paper are highlighted as follows: (1) We present E2E-MFD, a pioneering approach to efficient synchronous joint learning, innovatively integrating image fusion and object detection into a single-stage, end-to-end framework. This methodology significantly enhances the outcomes of both tasks. (2) We introduce a novel GMTA technique, designed to evaluate and quantify the impacts of the image fusion and object detection tasks. This aids in optimizing the training process's stability and ensures convergence to an optimal configuration of fusion detection weights. (3) Through comprehensive experimentation on image fusion and object detection, we demonstrate the efficacy and robustness of our proposed method.

## 2  Related Work

### 2.1  Multimodal Fusion Object Detection

Due to the powerful nonlinear fitting capabilities of deep neural networks, deep learning has made significant progress in low-level vision tasks, particularly in image fusion tasks [19; 20; 7; 21; 22; 23; 24]. Early efforts [9; 25; 13; 26; 27; 28] tended to achieve excellent fusion results by adjusting network structures or loss functions, overlooking the fact that image fusion should aim to improve the performance of downstream application tasks. Fusion images with good quality metrics may be suitable for human visual perception but may not be conducive to practical application tasks [16; 29]. Some research has acknowledged this issue, Yuan *et al.* [30] utilized various aligned modalities for improved oriented object detection [31; 32; 33; 34] to address the challenge of cross-modal weakly misalignment in aerial visible-infrared images. Liu *et al.* [17] proposing a joint learning method,

pioneered the exploration of the MF and OD combination methods. Then the optimization of the loss function for segmentation [16] and detection [14] has been validated to be effective in guiding the generation of fused images. They consider the downstream task network as an additional constraint to assist the MF network in generating fusion results with clearer objects. Zhao *et al.* [18] leveraged semantic information from OD features to aid MF and perform meta-feature embedding to generate meta-features from OD features, which are then used to guide the MF network in learning pixel-level semantic information. Liu *et al.* [15] proposed multi-interactive feature learning architecture for image fusion and segmentation by enhancing fine-grained mapping of all the vital information between two tasks, so that the modality or semantic features can be fully mutual-interactive. However, OD considers the semantic understanding of objects, while MF and segmentation primarily focus on the pixel-level relationship between image pairs. The optimization coupling of detection and fusion tasks becomes more challenging to investigate these complementary differences, from which both fusion and detection can benefit. It is worth mentioning that these visible-infrared multimodal fusion detection methods are usually designed in a cascaded structure with tedious training steps. Researchers lack the adoption of end-to-end architectures, which would enable one-step network inference to generate credible fused images and detection results through a set of network parameters.

## 2.2 Multi-task Learning

Multi-task learning (MTL) [35; 36] involves the simultaneous learning of multiple tasks through parameter sharing. Prior approaches involve manually crafting the architecture, wherein the bottom layers of a model are shared across tasks [37; 38]. Some approaches tailor the architecture based on task affinity [39], while others utilize techniques such as Neural Architecture Search [40; 37; 41] or routing networks [42] to autonomously discern sharing patterns and determine the architecture. An alternative method typically combines task-specific objectives into a weighted sum [43; 44; 45]. In addition, most approaches (e.g. [46; 47; 48; 49; 50]) aim to mitigate the effects of conflicting or dominating gradients. The approach of explicit gradient modulation [48; 49; 50; 51], has demonstrated superior performance in resolving conflicts between task gradients by substituting conflicting gradients with modified, non-conflicting gradients. Inspired by the above multimodal learning methods, we introduce a Gradient Matrix Task-Alignment method to align the orthogonal components contained in image fusion and object detection tasks thereby effectively eliminating the inherent optimization barrier that exists between two tasks.

# 3 The Proposed Method

## 3.1 Problem Formulation

MF-OD task concentrates on generating an image that benefits emphasizing objects with superior visual perception capability. The goal of OD is to find the location and identify the class of each object in an image which can naturally provide rich semantic information along with object location information. Therefore, the motivation of OD-aware MF is to construct a novel infrared and visible image fusion framework that can benefit from the semantic information and object location information contained in OD. For this purpose, we suppose a pair of visible image $x \in \mathbb{R}^{H \times W \times C_x}$ and infrared image $y \in \mathbb{R}^{H \times W \times C_y}$. The optimization model is formulated as:

$$\min_{\boldsymbol{\theta}_\mathrm{t}} \mathcal{L}\left(\boldsymbol{t}, \mathcal{N}\left(\boldsymbol{x}, \boldsymbol{y}; \boldsymbol{\theta}_\mathrm{t}\right)\right), \tag{1}$$

where $t$ represents the output of the different task network $\mathcal{N}$ with the learnable parameters $\boldsymbol{\theta}_\mathrm{t}$. $\mathcal{L}(\cdot)$ is a constraint term to optimize the network. Previous approaches solely design image fusion or object detection networks in a cascaded way, which can only achieve outstanding results for one task. To produce visually appealing fused images alongside accurate object detection results, we jointly integrate the two tasks into a unified goal synchronously which can be rewritten as:

$$\boldsymbol{\theta}_\mathrm{u}, \boldsymbol{\theta}_\mathrm{d} = \arg\min \omega \mathcal{L}^\mathrm{u}\left(\boldsymbol{u}, \Phi\left(\boldsymbol{x}, \boldsymbol{y}; \boldsymbol{\theta}_\mathrm{u}\right)\right) + (1 - \omega)\mathcal{L}^\mathrm{d}\left(\boldsymbol{d}, \Psi\left(\boldsymbol{x}, \boldsymbol{y}; \boldsymbol{\theta}_\mathrm{d}\right)\right) + \mathcal{S}\left(\boldsymbol{\theta}^\star\right), \tag{2}$$

where $\boldsymbol{\theta}^\star = \boldsymbol{\theta}_\mathrm{u}^s = \boldsymbol{\theta}_\mathrm{d}^s$, defined as the shared parameters for MF and OD networks. $u$ and $d$ denote the fused image and detection result, which are produced by the MF network $\Phi(\cdot)$ and OD network $\Psi(\cdot)$ with the learnable parameters $\boldsymbol{\theta}_\mathrm{u}$ and $\boldsymbol{\theta}_\mathrm{d}$. $w$ is a predefined weighting factor to balance the task training. $\mathcal{S}(\cdot)$ is a constrained term to jointly optimize the two tasks. In this paper, we regard the $\mathcal{S}(\cdot)$ as a feature learning constrained manner and achieve this goal by designing a Gradient Matrix Task-Alignment training scheme.

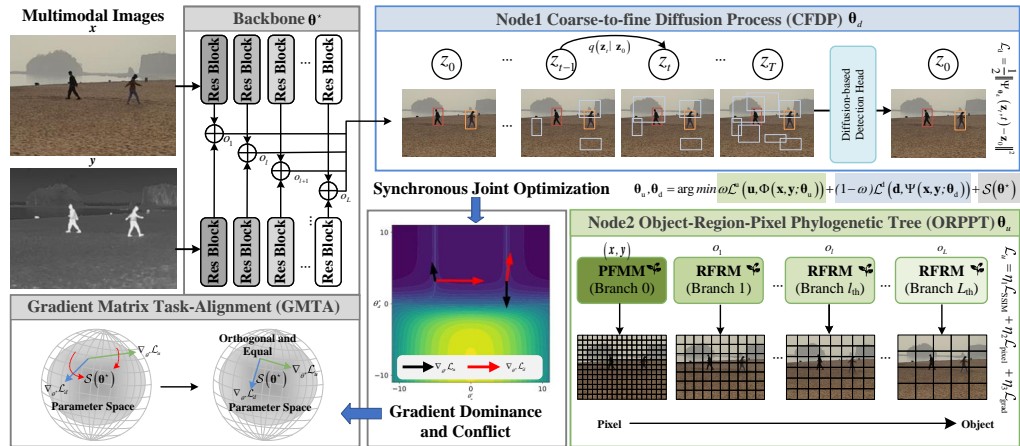

Figure 2: An overview of the proposed E2E-MFD framework, which consists of a backbone, nodes, and branches. The backbone is utilized to extract multimodal image features. A fine-grained fusion network (ORPPT) and diffusion-based object detection network (CFDP) are optimized by synchronous joint optimization (GMTA) in an end-to-end manner.

## 3.2 Architecture

Our proposed E2E-MFD is designed with parallel principle, composited by image fusion and object detection sub-network. Details of the whole architecture are shown in Figure 2. A fusion network (ORPPT) and object detection network (CFDP) can sufficiently realize the granularity-aware detail information and semantic information extraction.

**Object-Region-Pixel Phylogenetic Tree.** The important fact of that humans pay more attention to different regions from coarse to fine for object detection in object scale and image fusion in pixel scale. Inspired by a phylogenetic tree, to simulate humans to study the interactions of hierarchies under different granularity views, we construct an Object-Region-Pixel Phylogenetic Tree (ORPPT) as $\Phi(\cdot)$ to extract different features in multiple region scales. Given an image pair $\boldsymbol{x} \in \mathbb{R}^{W \times H \times C_x}$, $\boldsymbol{y} \in \mathbb{R}^{W \times H \times C_y}$, we firstly extract image features $f(\boldsymbol{x})$ and $f(\boldsymbol{y})$ by shared parallel backbone to save memory and computing resources. Then these features are added in channel dimension to obtain the final $L$ multimodal image features $\boldsymbol{o_1}, ..., \boldsymbol{o_l}, ..., \boldsymbol{o_L} \in \mathbb{R}^{W_1 \times H_1 \times C_1}$. The parameters in $f(\cdot)$ are shared with the OD network.

Although $f(\boldsymbol{x})$ and $f(\boldsymbol{y})$ can describe the characteristics of visible and infrared modalities, they lack insights into the multi-granularity perspective. Therefore, we utilize the branches including the one pixel feature mining module (PFMM) and $L$ region feature refine module (RFRM) to mine the multiple granularities from coarse to fine. The PFMM $B_0$ is set the same with feature fusion block in the MetaFusion [18] with the input pair $\boldsymbol{x}, \boldsymbol{y}$. We set $1, 2, \ldots, l, \ldots L$ to denote each region branch. For branch $l$, a CNN $\varphi_l(\cdot)$ is firstly utilized to extract the granularity-wise feature $\varphi_l(\boldsymbol{o_l}) \in \mathbb{R}^{W_2 \times H_2 \times C_2}$ at the region level. A set of learnable region prompts $\boldsymbol{R}_l = \left\{ \boldsymbol{r}_{l,m} \in \mathbb{R}^{C_2} \right\}_{m=1}^{M_l}$ are introduced to define the $M_l$ different regions of granularity-wise feature, where $r_{l,m}$ denotes the $m^{\text{th}}$ region prompt at branch $l$. Then the feature vector is mapped into the region mask $\boldsymbol{A}_l = \left\{ \boldsymbol{a}_{l,m} \in \mathbb{R}^{W_2 \times H_2} \right\}_{m=1}^{M_l}$ by conducting the dot product between the feature vector and region prompt followed by batch normalization and a ReLU activation:

$$\boldsymbol{A}_l = \text{ReLU}(\text{BN}(\boldsymbol{R}_l \cdot \varphi_l(\boldsymbol{o_l}))). \tag{3}$$

Finally, the vector of the object-level feature is weighted by the region mask and further aggregated to form region representation:

$$b_{l,m}^{i,j}(\boldsymbol{o_l}) = \boldsymbol{a}_{l,m}^{i,j} \varphi_l^{i,j}(\boldsymbol{o_l}), \tag{4}$$

where $b_{l,m}(\boldsymbol{o_l})$ denotes the $m^{\text{th}}$ region representation and $(i, j)$ denotes the spatial location. These region-level representations are further concatenated to form the observation $B_l(\boldsymbol{o_l}) = [b_{l,1}(\boldsymbol{o_l}), b_{l,2}(\boldsymbol{o_l}), \ldots, b_{l,M_l}(\boldsymbol{o_l})]$ of branch $l$. These multi-grained attentions concentrate operation on the spatial location information and the extent of the regions which are similar to the task requirements. The $B_1, B_2, \ldots, B_L$ are up-sampled to keep the consistent spatial size with the

pixel-level fusion features $B_0$. Then, the region-level fusion features $B_1, B_2, ..., B_L$ are assembled by addition operation followed by a Convolution with $1 \times 1$ kernel + ReLU to reduce the channel numbers. Highlighted region-level features are extracted by a Convolution with $1 \times 1$ kernel + Sigmoid and then injected into pixel-level features by multiplication and addition. Finally, five Convolutions with $3 \times 3$ kernel + ReLU layers are constructed to reconstruct the fusion result $u$.

**Coarse-to-Fine Diffusion Process.** Diffusion models, inspired by nonequilibrium thermodynamics, are a class of likelihood-based models. DiffusionDet [52] is the first neural network model to utilize the diffusion model for object detection, introducing a novel paradigm that achieves promising results compared to traditional object detection models. Combining diffusion simulation with the diffusion and recovery process of the object box, the Coarse-to-Fine Diffusion Process (CFDP) introduces it as an efficient detection head to assist fusion networks to focus more on object areas. CFDP model defines a Markovian chain of diffusion forward process by gradually adding noise to a set of bounding boxes. The forward noise process is defined as:

$$q\left(\boldsymbol{z}_t \mid \boldsymbol{z}_0\right) = \mathcal{N}\left(\boldsymbol{z}_t \mid \sqrt{\bar{\alpha}_t}\boldsymbol{z}_0, \left(1 - \bar{\alpha}_t\right)\boldsymbol{I}\right), \tag{5}$$

which transforms bounding boxes $z_0 \in \mathbb{R}^{N \times 4}$ to a latent noisy bounding boxes $\boldsymbol{z}_t$ for $t \in \{0, 1, \ldots, T\}$ by adding noise to $\boldsymbol{z}_0$. $\bar{\alpha}_t := \prod_{s=0}^{t} \alpha_s = \prod_{s=0}^{t}\left(1 - \beta_s\right)$ and $\beta_s$ represents the noise variance schedule. During training stage, a neural network $\Psi_{\boldsymbol{\theta}_d}\left(z_t, t\right)$ is trained to predict $z_0$ from $z_t$ by minimizing the training objective with $\ell_2$ loss:

$$\mathcal{L}_d = \frac{1}{2}\left\|\Psi_{\boldsymbol{\theta}_d}\left(\boldsymbol{z}_t, t\right) - \boldsymbol{z}_0\right\|^2. \tag{6}$$

At inference stage, bounding boxes $z_0$ is reconstructed from noise $z_T$ with the model $\Psi_{\boldsymbol{\theta}_d}$ and updating rule in an iterative way, i.e., $\boldsymbol{z}_T \to \boldsymbol{z}_{T-\Delta} \to \ldots \to \boldsymbol{z}_0$. In this work, we aim to solve the object detection task via the diffusion model. A neural network $\Psi_{\boldsymbol{\theta}_d}\left(\boldsymbol{z}_t, t, \boldsymbol{x}, \boldsymbol{y}\right)$ is trained to predict $z_0$ from noisy boxes $\boldsymbol{z}_t$, conditioned on the corresponding image pair $\boldsymbol{x}, \boldsymbol{y}$.

### 3.3 Loss Function

The total loss is combined with an image fusion loss function $\mathcal{L}_f$ and object detection loss $\mathcal{L}_d$. $\mathcal{L}_f$ consists of three types of losses, i.e., structure loss $\mathcal{L}_{SSIM}$, pixel loss $\mathcal{L}_{pixel}$ and gradient loss $\mathcal{L}_{grad}$. For one fused image, it should preserve overall structures and maintain a similar intensity distribution from source images. To this end, the structural similarity index (SSIM) is introduced in function:

$$\mathcal{L}_{SSIM} = \left(1 - SSIM(\boldsymbol{u}, \boldsymbol{x})\right)/2 + \left(1 - SSIM(\boldsymbol{u}, \boldsymbol{y})\right)/2, \tag{7}$$

where $\mathcal{L}_{SSIM}$ denotes structure similarity loss. In the fused image, we expect the object regions to have a more significant contrast compared to the background region. Therefore, the object regions need to preserve the maximum pixel intensity and the background region needs to be slightly below the maximum pixel intensity to bring out the contrast between the object and background. The ground-truth bounding boxes of the objects in images are denoted as $(x_c, y_c, w, h)$ for horizontal boxes and $(x_c, y_c, w, h, \theta)$ for rotated boxes, where $(x_c, y_c)$ is the center location, $w$ and $h$ are the width and height, $\theta = angle * pi/180$, respectively. Based on these ground-truth bounding boxes, we construct the object mask $I_m$, and the background mask is denoted as $1 - I_m$. The object regions pixel loss $\mathcal{L}_{pixel}^o$ and the background region pixel loss $\mathcal{L}_{pixel}^b$ are formulated as:

$$\mathcal{L}_{pixel}^o = \left\|I_m \circ \left(\boldsymbol{u} - max(\boldsymbol{x}, \boldsymbol{y})\right)\right\|_1, \mathcal{L}_{pixel}^b = \left\|(1 - I_m) \circ \left(\boldsymbol{u} - mean(\boldsymbol{x}, \boldsymbol{y})\right)\right\|_1, \tag{8}$$

where $\|\cdot\|_1$ stands for the $l_1$-norm. The operator $\circ$ denotes the elementwise multiplication, $max(\cdot)$ denotes element-wise maximization, and $mean(\cdot)$ denotes the element-wise average operation. Therefore, the object-aware pixel loss $\mathcal{L}_{pixel}$ is defined as:

$$\mathcal{L}_{pixel} = \mathcal{L}_{pixel}^o + \mathcal{L}_{pixel}^b. \tag{9}$$

Besides, gradient information of images always characterizes texture details, thus, we use $\mathcal{L}_{grad}$ to constrain these textual factors to a multi-scale manner:

$$\mathcal{L}_{grad} = \sum_{k=3,5,7}\left\|\nabla^k\boldsymbol{u} - max\left(\nabla^k\boldsymbol{x}, \nabla^k\boldsymbol{y}\right)\right\|_2^2, \tag{10}$$

where $\nabla$ denotes gradient operators that calculate by $\nabla = \boldsymbol{u} - \mathcal{G}(\boldsymbol{u})$ with combination of different Gauss ($\mathcal{G}$) kernel size $k$. Totally, we obtain $\mathcal{L}_u = \eta_1\mathcal{L}_{SSIM} + \eta_2\mathcal{L}_{pixel} + \eta_3\mathcal{L}_{grad}$.

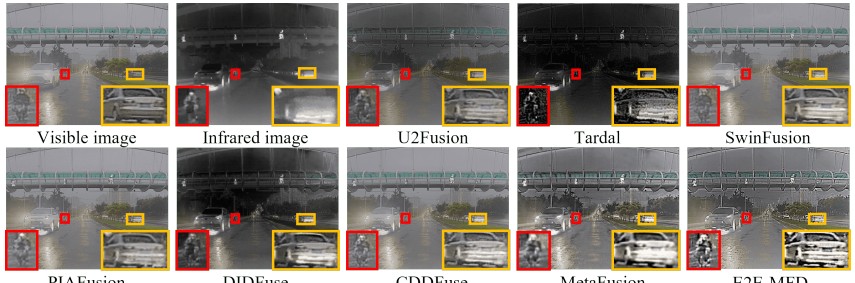

Figure 3: Visual results of image fusion on M3FD.

## 3.4 Gradient Matrix Task-Alignment

The MF and OD tasks have distinct optimization objectives. MF primarily emphasizes capturing the pixel-level relationship between image pairs, while OD incorporates object semantics within the broader context of diverse scenes. An inherent optimization barrier exists between these two tasks. We observe that the prevailing challenges in multi-task learning are arguably task dominance and conflicting gradients. We introduce a Gradient Matrix Task-Alignment (GMTA) by presenting the condition number to mitigate the undesired effects of the optimization barrier in task-shared parameters $\theta^\star$ which are supposed to be balanced between the MF and OD tasks. The individual task gradients of MF and OD task are calculated by $g_u = \nabla_{\theta^\star} \mathcal{L}_u$ and $g_d = \nabla_{\theta^\star} \mathcal{L}_d$ in the training optimization process. The gradient matrix can be defined as $G = \{g_u, g_d\}$. In multi-task optimization, a cumulative gradient $g = Gw$ is a linear combination of task gradients and the stability of a linear system is measured by the condition number of its matrix in numerical analysis. Hence the stability of the gradient matrix is equal to the ratio of the maximum and minimum singular values (non-negative) of the corresponding matrix: $\kappa(G) = \frac{\sigma_{\max}}{\sigma_{\min}}$. Learning from the Aligned-MTL [51], the condition number is optimal ($\kappa(G) = 1$) if and only if the gradients are orthogonal and equal in magnitude which means that the system of gradients has no dominance or conflicts:

$$\kappa(G) = 1 \iff <g_u, g_d> = 1. \tag{11}$$

The final linear system of gradients defined by $\hat{G}$ satisfies the optimal condition in terms of a condition number. Thereby, we consider the feature learning constraint $\mathcal{S}(\theta^\star)$ can be defined as the following optimization to eliminate instability in the training process:

$$\min_{\hat{G}} \|G - \hat{G}\|_F^2 \quad \text{s.t.} \quad \kappa(\hat{G}) = 1 \iff \min_{\hat{G}} \|G - \hat{G}\|_F^2 \quad \text{s.t.} \quad \hat{G}^\top \hat{G} = I. \tag{12}$$

The problem can be treated as a Procrustes problem and can be solved by performing a singular value decomposition (SVD) to $G$ ($G = U\Sigma V^T$) and rescaling singular values corresponding to principal components so that they are equal to the smallest singular value:

$$\hat{G} = \sigma U V^\top = \sigma G V \Sigma^{-1} V^T, \tag{13}$$

where,

$$(V, \lambda) = eigh(G^\top G), \tag{14}$$

$$\Sigma^{-1} = diag(\sqrt{1/\lambda_{max}}, \sqrt{1/\lambda_{min}}), \tag{15}$$

$eigh$ represents a function for finding eigenvectors $V$ and eigenvalues $\lambda$ and $diag$ stands for diagonal matrix. $\lambda_{max}$ and $\lambda_{min}$ are maximum eigenvalues and minimum eigenvalues from $\lambda$.The stability criterion, a condition number, defines a linear system to an arbitrary position scale. To alleviate this ambiguity, we choose the largest scale that guarantees convergence to the optimum: this is a minimal singular value of an initial gradient matrix:

$$\sigma = \sigma_{\min}(G) = \sqrt{\lambda_{min}}. \tag{16}$$

Table 1: Quantitative results of different fusion methods on TNO, RoadScene, and M3FD datasets. The model training (Tr.) and test (Te.) time is counted on an NVIDIA GeForce RTX 3090. The best result is **highlighted**.

| Task | Method | M3FD EN | M3FD MI | M3FD VIF | TNO EN | TNO MI | TNO VIF | RoadScene EN | RoadScene MI | RoadScene VIF | Tr. Time | Te. Time |
|---|---|---|---|---|---|---|---|---|---|---|---|---|
| MF | DIDFuse[13] | 6.13 | 14.65 | 1.51 | 6.30 | 15.30 | 1.47 | 6.67 | 16.65 | 1.55 | 3h9m38s | 0.096s |
| | U2Fusion[26] | 5.66 | 14.22 | 1.50 | 5.78 | 14.89 | 1.49 | 6.25 | 16.30 | 1.57 | 4h8m36s | 2.091s |
| | PIAFusion[58] | 5.75 | 13.92 | 1.59 | 5.05 | 13.61 | 1.36 | 6.37 | 16.22 | 1.58 | 5h35m20s | 0.003s |
| | SwinFusion[59] | 5.80 | 13.83 | 1.58 | 6.09 | 14.28 | 1.55 | 6.30 | 15.93 | 1.60 | 3h38m5s | 0.044s |
| | CDDFuse[12] | 5.77 | 13.82 | 1.58 | 6.21 | 15.03 | 1.49 | 6.54 | 16.54 | 1.57 | 5h59m59s | 0.096s |
| MF-OD | Tardal[17] | 5.72 | 14.68 | 1.47 | 5.87 | 14.99 | 1.43 | 6.72 | 16.98 | 1.54 | 5h36m28s | 0.093s |
| | Metafusion[18] | 6.20 | 15.19 | 1.54 | 6.29 | 16.03 | 1.44 | 6.35 | 16.76 | 1.57 | 6h47m38s | **0.002s** |
| | E2E-MFD | **6.36** | **15.47** | **1.65** | **6.40** | **16.28** | **1.60** | **6.79** | **17.11** | **1.69** | **2h50m32s** | 0.014s |

## 4 Experiments and Analysis

### 4.1 Dataset and Implementation Details

We conduct experiments on four widely-used visible-infrared image datasets: **TNO** [53], **RoadScene** [26], **M3FD** [17] and **DroneVehicle** [3]. TNO and RoadScene are just used to evaluate MF performance. M3FD is adopted to evaluate both MF and OD performance. RoadScene with 37 image pairs, TNO with 42 image pairs and M3FD with 300 pairs are only used for the MF task in the testing stage, and the MF network is trained by the M3FD dataset which is divided into a training set (2,940 image pairs) and a testing set (1,260 image pairs). Besides, DroneVehicle consists of 28,439 image pairs is utilized to train and test MF and OD for oriented objects. We conduct all the experiments with one GeForce RTX 3090 GPU, and the code of M3FD is based on Detectron2 [54], while the code of DroneVehicle is based on MMDetection 2.26.0 [55] and MMRotate 0.3.4 [56]. On the M3FD dataset, the pretrained DiffusionDet is used for the initialization of the OD network. In the training phase, E2E-MFD is optimized by AdamW with a batch size of 1. We set the learning rate to $2.5e - 5$ and the weight decay as $1e - 4$. The default training iteration is only 15,000. On the DroneVehicle dataset, the pretrained LSKNet [57] is used for the initialization of the object detection network, and we fine-tune it for 12 epochs with a batch size of 4. The E2E-MFD is optimized by AdamW and the learning rate and the weight decay is set to $1e - 4$ and 0.05.

### 4.2 Main Results

**Results on Multimodal Image Fusion.** Qualitative results of different fusion methods are depicted in Figure 3. All the fusion methods can fuse the main features of the infrared and visible images to some extent and we can observe two remarkable advantages of our method. First, the significant characteristics of infrared images can be effectively highlighted by our method. Our M3FD fusion image captures the person riding a motorcycle. In comparison with other methods, our method demonstrates high contrast and recognition of the objects. Second, our method preserves rich details from the visible images, including color and texture. Our advantages are evident in the fusion images across the M3FD dataset, such as the clear outline of the white car's rear and a man on a motorcycle. While retaining a substantial amount of detail, our method maintains a high resolution without introducing blurriness. In contrast, other methods fail to achieve these two advantages simultaneously. Sequentially, we provide quantitative results of different fusion methods in Table 1. Our E2E-MFD generally achieves the best metric values. Specifically, the largest average value on MI proves that our method transfers more considerable information from both source images. Values of EN reveal that our results contain edged details and the highest contrast between objects and the background. The large VIF shows our fusion results have high-quality visual effects and small distortion compared with the source images. Moreover, our method achieves the fastest training time to finish the joint learning at one stage which means that faster iterative updates can be done on new datasets. The test time to generate a fused image ranked third.

**Results on Multimodal Object Detection.** To more effectively evaluate the fusion images and observe their impact on downstream detection tasks, we conduct tests using the baseline detector YOLOv5s on all SOTA methods on the M3FD dataset. We follow the same parameter settings, and the visualization results are shown in Figure 4. The detection results are poor when using only single-modal image inputs, with instances of missed detection, such as the motorcycle and rider next to the car and people on the far right in the image. Almost all fusion methods reduce

Table 2: Quantitative results of object detection on M3FD dataset among all the image fusion methods + detector (i.e. YOLOv5s [60]). $^\star$ means using the fusion images generated by E2E-MFD for object detection training. The best result is **highlighted**.

| Task | Method | People | Car | Bus | Motorcycle | Lamp | Truck | mAP$_{50}$ | mAP$_{50:95}$ |
|---|---|---|---|---|---|---|---|---|---|
| V/I | Infrared | 49.3 | 67.1 | 72.9 | 35.8 | 43.6 | 61.6 | 85.3 | 55.1 |
| | Visible | 38.1 | 69.4 | 75.5 | 44.4 | 44.8 | 63.2 | 86.3 | 55.9 |
| MF | DIDFusion[13] | 45.8 | 68.8 | 73.6 | 42.2 | 43.7 | 61.5 | 86.2 | 56.2 |
| | U2Fusion[26] | 47.7 | **70.1** | 73.2 | 43.2 | 44.6 | 63.9 | 87.1 | 57.1 |
| | PIAFusion[58] | 46.5 | 69.6 | 75.1 | 45.4 | 44.8 | 61.7 | 87.3 | 57.2 |
| | SwinFusion[59] | 44.5 | 68.5 | 73.3 | 42.2 | 44.4 | 63.5 | 85.8 | 56.1 |
| | CDDFuse[12] | 46.1 | 69.7 | 74.2 | 42.2 | 44.2 | 62.7 | 87.0 | 56.5 |
| E2E-OD | CFT[61] | 52.0 | 68.2 | 79.2 | 49.9 | 45.2 | 69.6 | 89.8 | 60.7 |
| | ICAFusion[62] | 48.8 | 68.5 | 72.3 | 45.5 | 43.6 | 64.7 | 87.4 | 57.2 |
| MF-OD | Tardal[17] | 49.8 | 65.4 | 69.5 | 46.6 | 43.7 | 61.1 | 86.0 | 56.0 |
| | MetaFusion[18] | 48.4 | 66.7 | 70.5 | 49.1 | 46.4 | 59.9 | 86.7 | 56.8 |
| | Ours (YOLOv5s)$^\star$ | 51.0 | 67.9 | 69.4 | 50.2 | **48.7** | 61.6 | 87.9 | 58.1 |
| | Ours (DiffusionDet)$^\star$ | 58.5 | 67.7 | 79.9 | 50.3 | 46.2 | 70.2 | 90.3 | 62.1 |
| | Ours (E2E-MFD) | **60.1** | 69.5 | **81.4** | **52.2** | 47.6 | **72.2** | **91.8** | **63.8** |

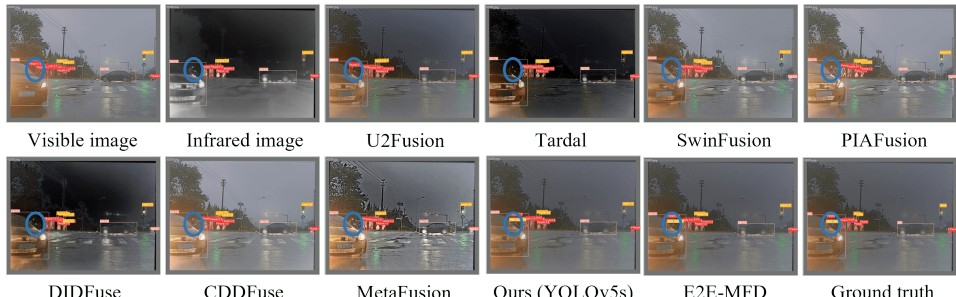

Visible image    Infrared image    U2Fusion    Tardal    SwinFusion    PIAFusion

DIDFuse    CDDFuse    MetaFusion    Ours (YOLOv5s)    E2E-MFD    Ground truth

Figure 4: Visual results of object detection on M3FD.

missed detection and improve confidence by fusing information from both modalities. Through the design of an end-to-end fusion detection synchronous optimization strategy, we obtain fusion images that are visually and detection-friendly, especially for occluded and overlapping objects, as seen in the blue ellipse with the motorcycle and the overlapping people on the far right in the image. To further assess the quality of fusion images, we conduct a fair comparison between our method and the SOTA methods on YOLOv5s. As shown in Table 2, the MF methods demonstrate a performance improvement compared to single-modal detection, indicating that well-fused images can effectively assist downstream tasks. In contrast, the fusion image we generate has achieved the best performance on YOLOv5s. Additionally, the detection performance of fusion images on DiffusionDet is also impressive, albeit slightly lower than when optimizing fusion and detection tasks

Table 3: Quantitative results of object detection on DroneVehicle test sets among all the SOTA methods, where $^\star$ means using the fusion images generated by E2E-MFD for object detection training and testing. The best result is **highlighted**.

| Modality | Detectors | Car | Truck | Freight Car | Bus | Van | mAP$_{50}$ |
|---|---|---|---|---|---|---|---|
| RGB | RetinaNet-OBB [63] | 67.5 | 28.2 | 13.7 | 62.1 | 19.3 | 38.1 |
| | Faster R-CNN-OBB [64] | 67.9 | 38.6 | 26.3 | 67.0 | 23.2 | 44.6 |
| | Gliding Vertex [65] | 75.8 | 46.1 | 33.8 | 68.1 | 38.7 | 52.5 |
| | YOLOv5s-OBB [60] | 89.0 | 53.6 | 41.9 | 84.8 | 32.6 | 60.4 |
| | LSKNet-OBB [57] | 89.5 | 70.0 | 51.8 | 89.4 | 56.9 | 71.5 |
| IR | RetinaNet-OBB [63] | 79.9 | 32.8 | 28.1 | 67.3 | 16.4 | 44.9 |
| | Faster R-CNN-OBB [64] | 88.6 | 42.5 | 35.2 | 77.9 | 28.5 | 54.6 |
| | Gliding Vertex [65] | 89.2 | 59.7 | 43.0 | 78.8 | 43.9 | 62.9 |
| | YOLOv5s-OBB [60] | 95.6 | 57.2 | 47.5 | 89.4 | 35.2 | 65.0 |
| | LSKNet-OBB [57] | 90.3 | 73.3 | 57.8 | 89.2 | 53.2 | 72.8 |
| RGB+IR | UA-CMDet [3] | 87.5 | 60.7 | 46.8 | 87.1 | 38.0 | 64.0 |
| | TSFADet [30] | 89.2 | 72.0 | 54.2 | 88.1 | 48.8 | 70.4 |
| | CALNet [66] | 90.3 | 76.2 | 63.0 | 89.1 | 58.5 | 75.4 |
| | Ours (YOLOv5s-OBB)$^\star$ | **96.7** | 69.9 | 49.9 | **92.6** | 44.5 | 70.7 |
| | Ours (LSKNet-OBB)$^\star$ | 90.3 | 77.0 | 63.5 | 89.5 | 59.0 | 75.9 |
| | Ours (E2E-MFD) | 90.3 | **79.3** | **64.6** | 89.8 | **63.1** | **77.4** |

simultaneously with E2E-MFD. Thanks to the collaborative optimization of both tasks, the detection performance is further enhanced. Furthermore, even when compared to end-to-end object detection methods (E2E-OD), our approach demonstrates significant performance improvements. This better underscores the advantages of our training paradigm and the effectiveness of our method.

**Results on Multimodal Oriented Object Detection.** As shown in Table 3, our fusion detection synchronous optimization strategy achieves the highest accuracy. Furthermore, the outstanding detection performance on YOLOv5s-OBB [60] and LSKNet using the generated fusion images (with at least 5.7% and 3.1% higher AP values compared to single modalities) demonstrate the robustness of our method. This validates the superior quality of the fusion images, indicating that they are not only visually appealing but also provide rich information for the detection task.

## 4.3 Ablation Studies

**Analysis of Gradient Matrix.** As described in Section 3.4, the MF and OD tasks pursue different optimization goals. To visualize the task dominance and conflicting gradients, we plot the gradient matrix in the training stage illustrated by Figure 5. We perform a GMTA operation every 1,000 iteration loss updates. Blue represents the gradients of shared parameters computed by the OD loss function, while yellow represents the gradients of shared parameters computed by the MF loss function. During the training process, it can be observed that the gradient values of the OD task are larger and dominant, while that of the MF task are smaller. This may affect the learning process of the fusion task during training. Conversely, the utilization of GMTA effectively mitigates this gradient dominance and conflict, facilitating a balance of shared parameters between MF and OD.

**Effect of Gradient Matrix Task-Alignment.** To verify the effectiveness of GMTA, we compare separate optimizations for MF and OD, as well as joint optimization, w/o and w indicating whether to use GMTA. Specifically, MF represents using only $\mathcal{L}_u$ to optimize the fusion network, OD represents using only $\mathcal{L}_d$ to optimize the object detection network, and E2E-MFD represents simultaneous optimization of the fusion and detection networks using both $\mathcal{L}_u$ and $\mathcal{L}_d$ loss functions. The results, as shown in Table 4, indicate that methods incorporating GMTA optimization constraints with shared weights achieve the best results for both MF and OD. This is because MF primarily emphasizes capturing the pixel-level relationship between image pairs, while OD incorporates object semantics within the broader context of diverse scenes. Therefore, optimizing the entire network with shared loss functions may be influenced by local optimal solutions of individual tasks. The accuracy of E2E-MFD (w/o GMTA) shows a slight decrease compared to separately training the detection network. In contrast, GMTA orthogonalizes the gradients of shared parameters corresponding to the two tasks, allowing the joint network to converge to an optimal point with fusion detection weights.

Table 4: The validation of GMTA on M3FD.

| Task | EN | MI | VIF | mAP$_{50}$ | mAP$_{50:95}$ |
|---|---|---|---|---|---|
| MF | 6.09 | 14.90 | 1.48 | / | / |
| OD | / | / | / | 90.28 | 62.75 |
| E2E-MFD (w/o GMTA) | 6.12 | 14.70 | 1.39 | 90.05 | 62.60 |
| E2E-MFD (w GMTA) | **6.36** | **15.47** | **1.65** | **91.80** | **63.83** |

To compare the effectiveness of different Multi-task learning methods on our algorithm, we selected three robust MTL optimization methods. As shown in Table 5, all MTL methods addressed the conflict between the MF and OD tasks to varying degrees. By introducing the concept of GMTA, we achieved a better balance in the gradient optimization process between the two tasks, resulting in the best performance.

Table 5: Ablation of different MTL methods on M3FD.

| Method | EN | MI | VIF | mAP$_{50}$ | mAP$_{50:95}$ |
|---|---|---|---|---|---|
| E2E-MFD (w/o GMTA) | 6.12 | 14.70 | 1.39 | 90.05 | 62.60 |
| PCGrad[50] | 6.13 | 15.01 | 1.48 | 90.59 | 62.71 |
| CAGrad[47] | 6.17 | 15.05 | 1.48 | 90.71 | 62.74 |
| Nash-MTL[49] | 6.29 | 15.28 | 1.51 | 90.91 | 62.97 |
| E2E-MFD (w GMTA) | **6.36** | **15.47** | **1.65** | **91.80** | **63.83** |

The GMTA process operates during the computation and updating stages of two gradients. The GMTA is performed approximately every $n$ iteration (gradient update), focusing on balancing the independence and coherence of various tasks. Table 6 presents the ablation analysis of the $n$ parameter. Decreasing $n$ initially disrupts task optimization due to frequent alignment, while increasing $n$ becomes crucial when

Table 6: Ablation studies of the iteration parameter $n$ on M3FD dataset.

| $n$ | EN | MI | VIF | mAP$_{50}$ | mAP$_{50:95}$ |
|---|---|---|---|---|---|
| 500 | 5.93 | 14.78 | 1.58 | 90.93 | 62.73 |
| **1000** | **6.36** | **15.47** | **1.65** | **91.80** | **63.83** |
| 1500 | 6.24 | 15.08 | 1.62 | 91.10 | 62.96 |
| 2000 | 6.13 | 14.69 | 1.45 | 90.35 | 62.75 |

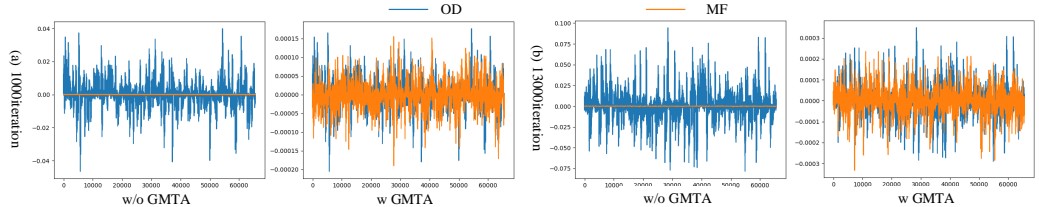

Figure 5: Visualization of task dominance and conflicting gradients in joint learning of OD and MF.

the network determines task optimization directions. However, excessively large $n$ leads to significant deviations in task paths, making alignment more challenging and negatively impacting performance.

**Study of Branches in the Object-Region-Pixel Phylogenetic Tree.** We investigate the combination of the pixel feature mining module (0) and the region feature refinement module (1,2,3,4). The results are shown in Table 7. It can be observed that with the increase in the number of branches, the fusion network achieves higher image fusion quality and object detection performance. Region features provide the fusion network with multi-level semantic features of objects. However, when higher-level semantic object information is added, the performance of the fusion network declines. This is because the detailed information contained in the deeper layers of the backbone structure shared with the detection network decreases abruptly, which may affect the fusion network's absorption of these features, thereby influencing pixel-level fusion tasks.

Table 7: Ablation study of the number of ORPPT branches on M3FD.

| Branch | EN | MI | VIF | mAP$_{50}$ | mAP$_{50:95}$ |
|---|---|---|---|---|---|
| 0 | 6.10 | 15.19 | 1.54 | 91.51 | 63.20 |
| 0,1 | 6.10 | 15.29 | 1.54 | 91.65 | 62.96 |
| 0,1,2 | 6.19 | 15.28 | 1.58 | 91.73 | 63.46 |
| 0,1,2,3 | **6.36** | **15.47** | **1.65** | **91.80** | **63.83** |
| 0,1,2,3,4 | 5.99 | 14.31 | 1.40 | 91.73 | 63.55 |

We have implemented the Object-Region-Pixel Phylogenetic Tree (ORPPT) to explore the hierarchical interactions under different granularity views and extract various features across multiple region scales, as introduced in Section 3.2. As shown in Figure 6, the detailed information will decrease as the backbone structure shared by the detection network deepens, resulting in a decrease in detailed information. This may affect the absorption of these features by the fusion network of the shared backbone network, thereby affecting pixel-level fusion tasks. This provides evidence for our analysis of the reasons for Section 4.3 ablation experiment for Object-Region-Pixel Phylogenetic Tree. This illustrates the importance of balancing the semantic information provided between the OD and MF network with pixel-level information.

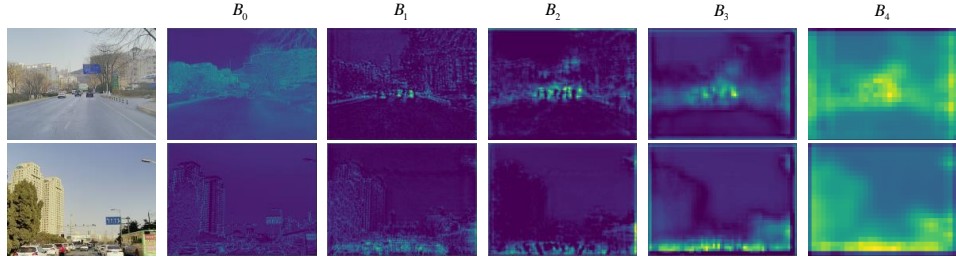

Figure 6: Feature map visualization of various branches in the ORPPT.

## 5 Conclusion

Within this paper, an end-to-end optimization is proposed to formulate fusion and detection in a harmonious one-stage training process. We introduce a object-region-pixel phylogenetic tree structure and coarse-to-fine diffusion process to simulate these two tasks in different visual perceptions needed for diverse task requirements. In addition, we align the orthogonal components of the fusion detection linear system of the gradients by gradient matrix task-alignment. By unrolling the model to a well-designed fusion network and diffusion-based detection network, we can generate a visual-friendly result for fusion and object detection in an efficient way without tedious training steps and inherent optimization barriers.

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

# A  Appendix

## A.1  Metric

Three metrics are used for MF evaluation: entropy (EN) [67], mutual information (MI) [68] and visual information fidelity (VIF) [69]. EN evaluates the information richness in an image, and the higher EN means more information. MI evaluates the information similarity between the input images and fused images. The higher MI illustrates more information of the input images is fused. VIF measures the ability to extract visible information from the input image, and a larger VIF represents less visible distortion in the fused result. Here, we use the V channel in the HSV space of the fusion results to calculate these metrics. Moreover, we use $mAP_{50:95}$ [70] to comprehensively evaluate OD performance, where the average of mAPs sampling every 5 from $AP_{50}$ to $AP_{95}$ is calculated. A higher $mAP_{50:95}$ means a better OD effect.

## A.2  Experiment setting on YOLOv5s

To more effectively evaluate the fusion images and observe their impact on downstream detection tasks, we conduct tests using the baseline detector YOLOv5s on all SOTA methods on the M3FD dataset. All images are resized to 1024×1024 and trained from scratch for 300 epochs with a batch size of 64. In addition, the experiments for DroneVehicle dataset are conducted on YOLOv5s-OBB using the generated fusion images, all images are resized to 640×640 and trained from scratch for 50 epochs with a batch size of 64.

## A.3  Experiments on DroneVehicle

DroneVehicle comprises aerial RGB-IR images captured by drones, encompassing various scenes from an aerial perspective. It contains five categories of target objects. The dataset consists of 28,439 pairs of images, divided into training, validation, and test sets. We train the MF network on the training set and perform inference directly on the test set. Finally, fusion images are generated for both the training and test sets. All detection accuracies are obtained on the test set. The visualization results of the fusion and detection of DroneVehicle are shown in Figure 7 and 8. In the process of generating fusion images, we first obtain RGB images, then convert them to HSV format, and only take the V channel as the final fusion image. It preserves the brightness information of the objects while containing rich details from visible images. As shown in Figure 7, the first row and the second row are images extracted from the training set and the test set, respectively. Although the style of the fusion images resembles that of the infrared images, by combining visible image information, the fusion images can differentiate different parts of each object, reflecting varying degrees of brightness information. This is advantageous for fine-grained classification. Additionally, as shown in Figure 8, the visually friendly fusion images also assist in the object detection task, allowing for sufficient detection even in cases of small and dense objects.

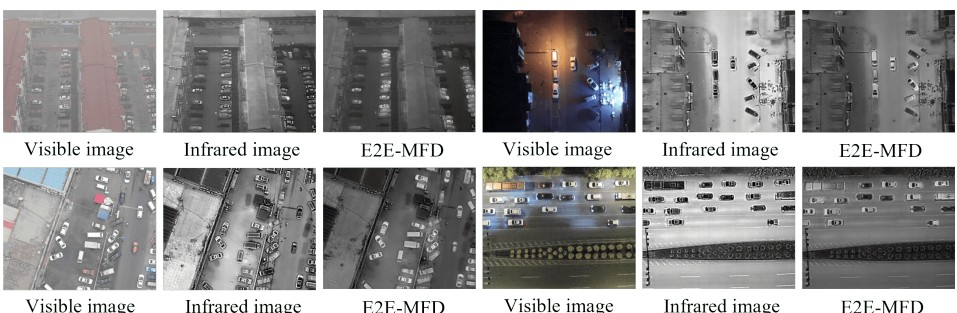

Figure 7: Qualitative results of image fusion on DroneVehicle.

## A.4  Experiments on Coase-to-Fine Diffusion Process

We conducted ablation experiments on CFDP in Table 8, investigating its inclusion and the number of proposed boxes. In the setting without CFDP, we maintained the backbone network while substituting

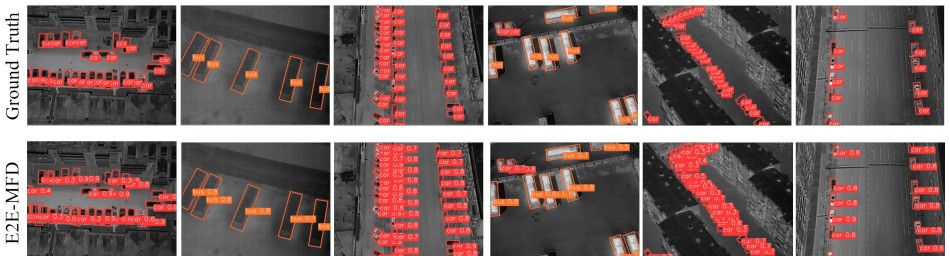

Figure 8: Visual results of object detection on DroneVehicle.

CFDP with RPN (Region Proposal Network), standard components in two-stage object detectors. Results indicate that CFDP enhances detailed information capture and precise box guidance, thereby enhancing fusion image quality and detection performance. For optimal balance between performance and efficiency, we selected 500 proposal boxes.

Table 8: Ablation studies of the CFDP on M3FD dataset.

| Settings | Proposal boxes | EN | MI | VIF | mAP$_{50}$ | mAP$_{50:95}$ | Tr.Time |
|----------|----------------|------|-------|------|-------|-------|----------|
| w/o CFDP | 500 | 5.71 | 14.39 | 1.45 | 90.13 | 61.98 | 2h52m11s |
| w CFDP | 300 | 6.01 | 14.57 | 1.53 | 90.89 | 63.29 | 2h23m45s |
| | 500 | 6.36 | **15.47** | **1.65** | 91.80 | **63.83** | 2h50m32s |
| | 1000 | **6.37** | 15.34 | 1.63 | **92.05** | 63.75 | 3h32m30s |

## A.5 More Fusion Visualization Results

More comparisons of infrared-visible image fusion visualization results are depicted in Figure 9, 10 and 11. These fusion results demonstrate the advantages of synchronously optimizing fusion and detection tasks. With minimal training costs, we obtain fusion images that are visually and detection-friendly. Specifically, the fusion images retain significant object information extracted from infrared images, while also preserving detailed information such as texture, color, and background from visible images. Our method effectively combines the strengths of both modalities to enhance the overall performance of the detection.

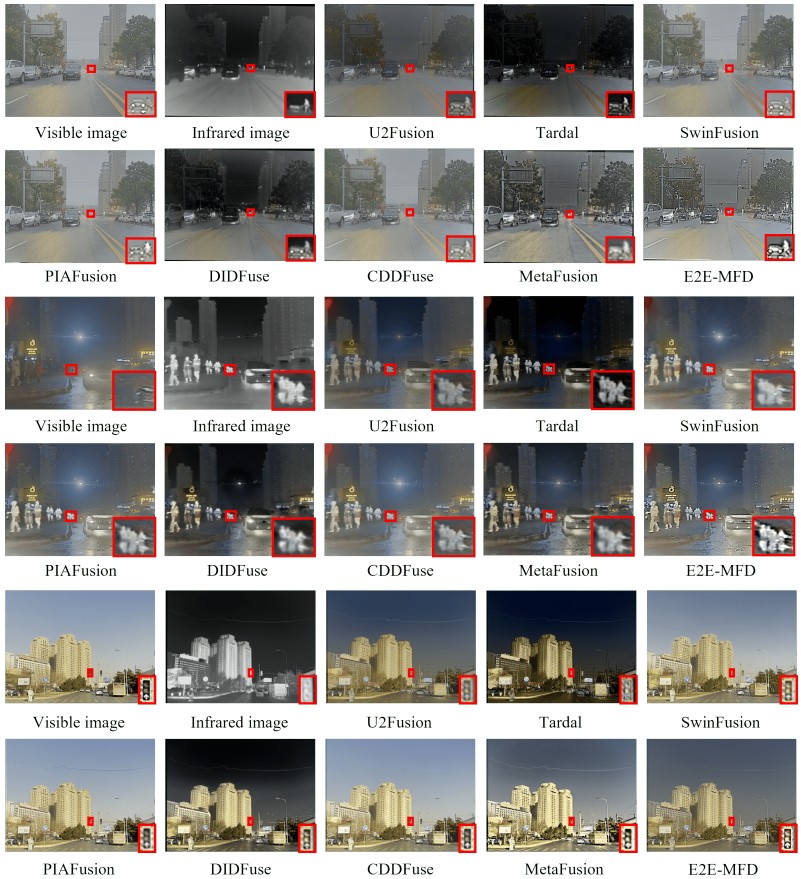

Figure 9: Qualitative results of image fusion on M3FD.

## A.6 More Detection Visualization Results

The visualization of infrared-visible object detection is demonstrated in Figure 12. Our fusion images are noticeably superior due to the effective combination of object information from infrared images and texture details from visible images. This integration enables the object detection network to distinguish between the background and the object clearly. Additionally, for small and occluded objects, the clear edge details assist the network in achieving improved detection results.

## A.7 Limitations and Broader Impacts

**Limitations.** Our E2E-MFD approach is effective for the joint learning of the multimodal fusion and object detection task and has been validated on various datasets. However, the validations of the current model rely on the visible and infrared modalities. Constrained by the scarcity of relevant datasets within the community, the paper lacks validation with additional datasets containing new modalities. Future research will focus on addressing this gap by exploring, constructing, and incorporating more diverse multimodal datasets serving multimodal fusion detection.

**Broader Impacts.** Our paper aims to broaden the applicability of joint learning of multimodal data and object detection to various research domains. However, this broader scope may present challenges when using the model in domains that include harmful content. These challenges arise from the data itself, rather than from the model. Therefore, it is crucial to have adequate data regularization to effectively address these concerns.

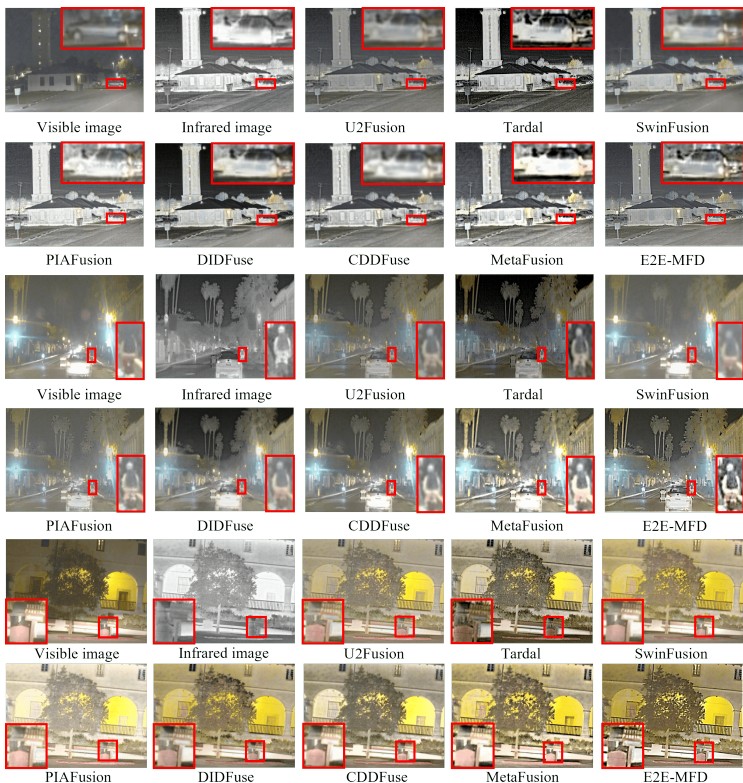

Figure 10: Qualitative results of image fusion on RoadScene.

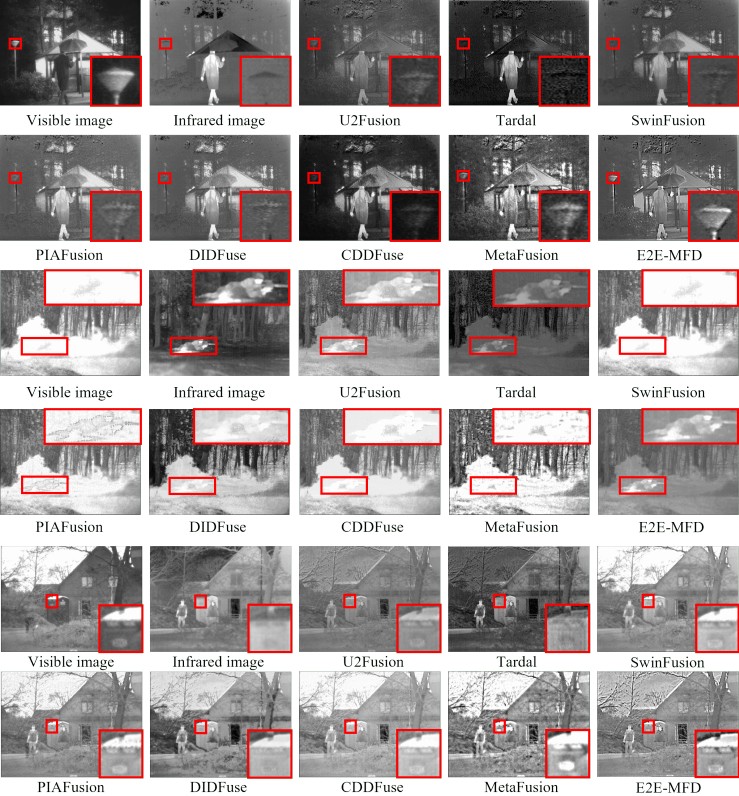

Figure 11: Qualitative results of image fusion on TNO.

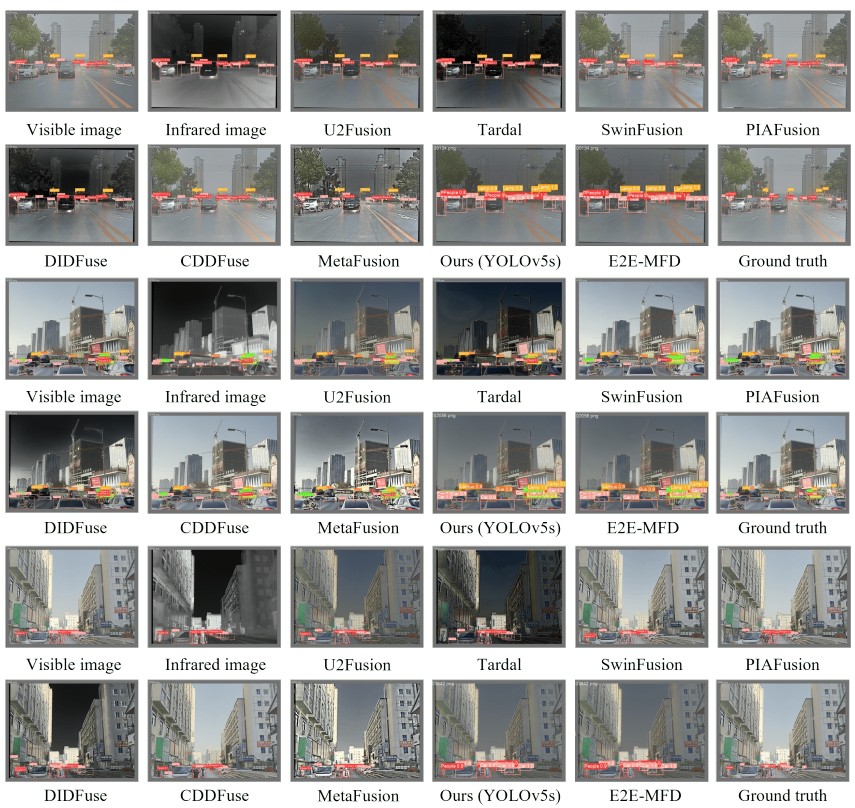

Figure 12: Visual results of object detection on M3FD.

