# OpenReview forum: "E2E-MFD: Towards End-to-End Synchronous Multimodal Fusion Detection"
_NeurIPS.cc/2024/Conference — NeurIPS 2024 oral_

### Official Review · Reviewer_cEzg · 2024-07-04

**Soundness:** 4
**Presentation:** 4
**Contribution:** 4
**Rating:** 8
**Confidence:** 5

**Summary:**

This paper introduces a novel end-to-end algorithm named E2E-MFD for multimodal image fusion and object detection. Unlike existing joint learning methods, its key innovation lies in the synchronous joint optimization approach, simplifying the fusion detection process into a single training step and enhancing efficiency compared to traditional multi-step methods. To harmonize the losses between the image fusion and object detection networks, a Gradient Matrix Task-Alignment method is proposed. This method balances the gradients of shared parameters between the image fusion and object detection tasks, addressing the challenges of task dominance and conflicting gradients in multi-task learning. Additionally, an image fusion network with an Object-Region-Pixel Phylogenetic Tree is designed to perceive information at different granularity levels. Experimental results demonstrate the performance of the proposed method in both image fusion and object detection.

**Strengths:**

- The idea of learning image fusion and object detection tasks simultaneously to mutually benefit each other is intriguing and reasonable.

- An end-to-end fusion detection algorithm is proposed, effectively avoiding the local optimum problem encountered in multi-stage training models. Specific modules such as Gradient Matrix Task-Alignment and Object-Region-Pixel Phylogenetic Tree are introduced to achieve this goal.

- Sufficient Experiments demonstrate that these modules facilitate the learning process, and jointly optimizing these two tasks outperforms other existing pipelines.

- The authors clearly describe their methods in the paper and enhance its comprehensibility through the judicious use of formulas and figures.

**Weaknesses:**

The overall idea is pretty interesting and reasonable. I can see the insight in the proposed method. However, there are some typos in paper:
1.The L_SSIM in the line 179 seems not same with the Equation (7).
2.I see a dashed arrow in the figure 1, what’s this mean?

**Questions:**

1.In "Study of branches in the Object-Region-Pixel Phylogenetic Tree", the authors analyzed the reasons for performance degradation under settings 0, 1, 2, 3, and 4, and provided visual evidence. Could you elaborate on why the fusion performance only starts to decline after adding the fourth setting?
2.For Figure 4, the targets are only circled. Enlarging the highlighted areas, similar to the other figures, would make them clearer.

**Limitations:**

Expanding new modal datasets or implementing modality conversion between multimodal data will become a solution to the single issue raised in the paper about publicly available multimodal object detection datasets.

---

> ### Author Rebuttal · Authors · 2024-08-03
>
> # Reviewer cEzg
> Thank you for your feedback.
>
> **A1**: Regarding the typo in Equation (7) and the inconsistency with $\mathcal{L}_{\text {SSIM }}$ on line 179, we apologize for the oversight. We have revised line 179 to ensure consistency with the description in Equation (7).
>
> **A2**: The dashed arrow in Figure 1 represents Dashed arrows indicating cutting off gradient flow. We have clarified this in the figure caption for better understanding.
>
> **A3**: In the study of branches in the Object-Region-Pixel Phylogenetic Tree, we analyzed the performance degradation under settings 0, 1, 2, 3, and 4, and provided visual evidence. The fusion performance begins to decline after adding the fourth setting primarily due to the increased complexity and interaction among multiple branches. As more branches are added, the network may struggle to effectively balance and integrate pixel-level and object-level information, leading to a gradual decline in fusion performance.
>
> **A4**: Regarding Figure 4, we acknowledge your suggestion to enlarge the highlighted areas where targets are circled. We have enhanced the clarity of these areas in Figure 4 to provide a clearer representation of the targets, similar to other figures in the manuscript.

---

> > ### Comment · Reviewer_cEzg · 2024-08-11
> >
> > The rebuttal has addressed my concerns. Due to the novel motivation, clear methodology, and comprehensive experimental analysis, I will maintain my score. I suggest incorporating these modifications into the paper.

---

> > > ### Author Response · Authors · 2024-08-13
> > > **Official Comment by Authors**
> > >
> > > Thank you for your prompt comments and for affirming our rebuttal. As you suggested, we will incorporate these explanations and revisions into the paper to enhance clarity.

---

### Official Review · Reviewer_6qWk · 2024-07-12

**Soundness:** 4
**Presentation:** 4
**Contribution:** 4
**Rating:** 8
**Confidence:** 5

**Summary:**

This paper focuses on the task of multimodal image fusion detection, combining texture details and target semantic information. An end-to-end multimodal fusion detection algorithm named E2E-MFD is proposed, which employs synchronous joint optimization, differing from existing independent or cascaded joint methods. The authors introduce a Gradient Matrix Task-Alignment method to help resolve gradient conflict issues in the image fusion and object detection tasks. Experiments on horizontal and oriented object detection datasets demonstrate the effectiveness of this method.

**Strengths:**

1. This paper presents the first attempt to achieve simultaneous single-stage training of image fusion and object detection, and the results appear very promising.

2. Inspired by multitask learning, the Gradient Matrix Task-Alignment method is introduced to reasonably balance the loss functions, thereby converging the fusion detection weights to optimal points.

3. The multi-granularity strategy in the Object-Region-Pixel Phylogenetic Tree demonstrates its effectiveness in learning shared parameters, thereby enhancing object detection performance.

4. The writing and figures in the paper are clear and easy to understand. Proper use of formulas enhances the comprehensibility of their method.

5. The experiments and ablation studies comprehensively demonstrate the results.

**Weaknesses:**

The paper presents a thorough and well-executed series of experiments that significantly contribute to the strength and credibility of the research. However, I think some problems need to be addressed:
1. In the experiments, YOLOv5s is compared. But why not comparing with the latest yolo?
2. The three backbone networks involved in the Figure 1 of the paper are not specified in the text, which would limit other researchers to know the details.
3. In the line 22, "a MF network" should be "an MF network".

**Questions:**

See weaknesses above.

**Limitations:**

The authors note that current model validation relies on visible light and infrared modalities due to limited relevant datasets within the community. They express a need for new dataset guidelines and contributions to the open-source community to address multi-modal dataset validation challenges in the future.

---

> ### Author Rebuttal · Authors · 2024-08-05
>
> # Reviewer 6qWk
> Thank you for your constructive insights.
>
> **A1**: We acknowledge your point regarding comparing YOLOv5s with the latest version of YOLO. We chose YOLOv5s as it represents a well-established baseline in the field, and our focus was on evaluating the results of the detection accuracy of different fusion methods under the same detector. It is consistent with papers "CVPR2023 MetaFusion: Infrared and Visible Image Fusion via Meta-Feature Embedding from Object Detection" and "CVPR2022 Target-aware Dual Adversarial Learning and a Multi-scenario Multi-Modality Benchmark to Fuse Infrared and Visible for Object Detection. However, we will consider including a comparison with the latest YOLO version in future work to provide a more comprehensive evaluation.
>
> **A2**: Thank you for pointing out this omission. We apologize for the oversight. In Figure 1, the three backbone networks represent the same backbone. The parameters included are defined as the shared parameters for MF and OD networks. We will clarify this in the revised manuscript to ensure that other researchers have the necessary details for replication and comparison purposes.
>
> **A3**: Thank you for your corrections; we have made the necessary revisions.

---

> > ### Comment · Reviewer_6qWk · 2024-08-13
> >
> > I would like to thank the authors for their response. My concerns have been solved, and I will keep my positive rating.

---

> > > ### Author Response · Authors · 2024-08-13
> > > **Official Comment by Authors**
> > >
> > > Thank you for your prompt comments and recognition of our paper.

---

### Official Review · Reviewer_G1Ro · 2024-07-12

**Soundness:** 4
**Presentation:** 4
**Contribution:** 4
**Rating:** 9
**Confidence:** 3

**Summary:**

This paper proposes a joint learning diagram for multimodal fusion and object detection with task alignment module.

The suggested network achieves SOTA performance with affordable computational cost.

**Strengths:**

This paper presents a novel approach to learning image fusion and objection detection in a synchronous and joint way

The proposed network achieves the SOTA performance in both tasks.

**Weaknesses:**

Globally, I am ok with the significance of the work with the SOTA performance, despite the fact that it comes with additional computational cost.

I am more concerned by other issues:

1. After reviewing, this paper gives me the impression that the proposed method is more like a combination of existing works/modules/tricks to achieve the SOTA performance. In other words, I am concerned by the novelty.

2. Secondly, this paper is hard to read and follow. The motivation of the proposed work is not strong enough compared to SOTA works. The diagrams are a little bit confusing. The writing needs to be improved.

3. I am ok with all the proposed modules and blocks to be claimed as novel, such as the blocks in nodes 1 and 2. It seems that the main contribution that the author claimed is on task alignment. This seems to be a very generic learning strategy. The authors should further validate its effectiveness with other works such as MetaFusion or other applications.

**Questions:**

n/a

**Limitations:**

See weakness

---

> ### Author Rebuttal · Authors · 2024-08-05
>
> # Reviewer G1Ro
>
> Thank you for your feedback.
>
> **A1**: Reviewer 6qWk and reviewer cEzg both affirmed the novelty of our paper. As reviewer cEzg commented, "The idea of simultaneously learning image fusion (MF) and object detection (OD) tasks to mutually benefit each other is intriguing and reasonable." **It is important to note that the contribution of this paper lies in introducing the first end-to-end joint training paradigm in the fusion detection field.** In current research, joint learning algorithms are an emerging hotspot, leveraging fusion networks and object detection synergies to enhance image informativeness and improve detection performance. However, **existing** optimization methods are typically **non-end-to-end**, relying on multiple steps, as shown in Figure 1. These approaches are cumbersome and reduce training efficiency. Therefore, we propose the **first end-to-end synchronous joint optimization** algorithm, E2E-MFD, which facilitates interaction between intrinsic features from both domains through synchronous joint optimization, enabling a **streamlined, one-stage process**.
>
> Moreover, our end-to-end solution does not merely concatenate two tasks (modules) but involves dedicated module design. To harmonize **fine-grained details** with **semantic information**, we introduce the novel concept of an Object-Region-Pixel Phylogenetic Tree (ORPPT) coupled with a coarse-to-fine diffusion processing (CFDP) mechanism. Additionally, we find that the MF and OD tasks have naturally distinct optimization objectives. MF primarily focuses on capturing pixel-level relationships between image pairs, while OD integrates object semantics within diverse scene contexts. Therefore, there exists an **inherent optimization barrier** between these two tasks. To address this, we especially introduce the concept of **gradient alignment** in the multi-task learning field, proposing GMTA to align gradients for o**ptimizing task dominance and resolving conflicting gradients** in the end-to-end joint training process. As you mentioned, we ultimately propose a novel approach to learning image fusion and objection detection synchronously and jointly. We are the first to innovatively **integrate image fusion and object detection** into a single-stage, end-to-end framework, achieving **SOTA results** on multiple datasets.
>
> **A2**: As pointed out by reviewer 6qWk, “This paper presents the first attempt to achieve simultaneous single-stage training of image fusion and object detection, and the results appear very promising. Inspired by multitask learning, the GMTA method is introduced to reasonably balance the loss functions, thereby converging the fusion detection weights to optimal points.” As discussed in A1, previous SOTA methods relied on non-end-to-end optimization approaches, dividing joint training into multiple steps, which led to complexity during training. These methods excessively emphasized leveraging OD information for MF enhancement, complicating parameter balancing and making them susceptible to local optima of individual tasks. **Therefore, achieving a unified feature set that simultaneously satisfies the characteristics of each task through end-to-end training remains a formidable challenge.** This paper introduces E2E-MFD, an end-to-end multimodal fusion detection algorithm. E2E-MFD aims to seamlessly integrate detailed image fusion and object detection from coarse to fine levels. We introduce the gradient alignment concept from the multi-task learning domain, aiming to eliminate conflicting gradients between object detection and multimodal fusion tasks through the design of GMTA optimization mode. **By facilitating synchronous joint optimization and fostering interaction between intrinsic features from both tasks, E2E-MFD achieves a streamlined single-stage process in an end-to-end manner.** In addition, reviewers bix9, 6qWk, and cEzg both acknowledged our writing presentation. We will strive to improve our writing skills to the best of our ability. Please help us identify which diagrams have caused you confusion. We will make every effort to revise the diagrams and provide comprehensive explanations.
>
> **A3**: Our goal is to explore an end-to-end joint training approach. Thank you for recognizing the **novelty** of the **design of node1 and node2**. In this framework, Node 1 serves the object detection task, while Node 2 acts as a module for MF tasks, with personalized settings harnessing the respective roles of the nodes. As you mentioned, one of our primary contributions is introducing **an end-to-end synchronous training paradigm** for multimodal fusion detection, where synchronized joint optimization allows both tasks to complement each other synergistically. This collaboration enables MF to generate richer, more informative images, enhancing the performance of OD, which in turn provides valuable semantic insights to MF for accurate localization and identification of objects within scenes. However, it is crucial to address the issue of **gradient conflicts** in the joint optimization of fusion and detection tasks, known as task consistency. This challenge, though a common concern in multitask learning, is effectively tackled for the **first time** in the realm of **multimodal fusion detection** through end-to-end training. As reviewer cEzg noted, “An end-to-end fusion detection algorithm is proposed, effectively avoiding the local optimum problem encountered in multi-stage training models.”
>
> Additionally, it should be noted that the training approach of Metafusion is not end-to-end. Our algorithm essentially represents an advanced end-to-end version of “metafusion,” (i.e., a non-end-to-end image fusion method) taken to its ultimate level of refinement.

---

> > ### Comment · Reviewer_G1Ro · 2024-08-13
> >
> > I thank the authors for the rebuttal. I don't have other questions.

---

> > > ### Author Response · Authors · 2024-08-13
> > > **To Reviewer G1Ro**
> > >
> > > Thank you for taking the time to reply. We are pleased to hear that we have addressed your concerns. If you have any further questions, please let us know promptly so that we can resolve them in the remaining time. We hope you will reconsider our score. Thank you again.

---

### Official Review · Reviewer_bix9 · 2024-07-13

**Soundness:** 2
**Presentation:** 3
**Contribution:** 2
**Rating:** 4
**Confidence:** 5

**Summary:**

This paper proposed an end-to-end algorithm for multimodal fusion detection, experiments on fusion and detection tasks showed the better performance than some methods.

**Strengths:**

This paper proposed an end-to-end algorithm with one-stage training process，for multimodal fusion detection, experiments on fusion and detection tasks showed the better performance than some methods.

**Weaknesses:**

1. Why use the V channel in the HSV space of the fusion results to calculate the metrics?
2. The best result of car detection highlighted in table 2 is wrong. Additionally, add analysis of why the proposed algorithm couldn’t realize the best detection effect of car.
3. Provide a detailed justification for the chosen datasets. Explaining why these specific datasets are representative or challenging.
4. The details of GMTA should be added.
5. It is mentioned that the GMTA operation is executed every 1000 iterations, but more specific implementation details, such as the specific setting and selection basis of parameters, are lacking.
6. Ablation studies for CFDP should be added to verify how CFDP impacts the final results.
7. The advantages of ORPPT and GMTA compared with existing techniques are not fully demonstrated. It needs to be more explicit about how these innovations solve existing problems or lead to performance improvements.
8. More SOTA methods and metrics should be added for image fusion task.

**Questions:**

See the weaknesses.

**Limitations:**

See the weaknesses.

---

> ### Author Rebuttal · Authors · 2024-08-05
>
> # Reviewer bix9
> Thanks for your comments.
>
> **A1:** This is a common default setting in the field such as "CVPR23 MetaFusion". V (brightness) channel can effectively measure the algorithm’s ability to handle low-light environments.
>
> **A2:** We have made revision in Tab. 2. It's common to observe fluctuations in detection accuracy within a single category on M3FD dataset in Tab. R.1. Despite fluctuations, our algorithm significantly outperforms others in $\text{mAP}$.
>
> **A3:** We utilize widely recognized datasets with diverse tasks, sufficient quantity, and complex environment. TNO and Roadscene are evaluation datasets for MF. TNO captures multispectral images day and night, while Roadscene comprises aligned image pairs from road scenes with vehicles and pedestrians. We also enrich experimental validation by integrating horizontal and oriented OD datasets. The M3FD dataset covers complex scenarios with diverse camera angles. Additionally, DroneVehicle dataset is considered for oriented OD with diverse scenes from an aerial viewpoint.
>
> **A4:** The process of GMTA is illustrated in Section 3.4 in the paper and we detailed this section to eliminate the confusion. GMTA mitigates the undesired effects of the optimization barrier in task-shared parameters which are supposed to be balanced between the MF and OD tasks. Conflicts in the gradient matrix $\boldsymbol{G}$ are related to a negative cosine distance between gradient vectors ($<\boldsymbol{g}_u,\boldsymbol{g}_d><0$), while dominance is caused by an imbalance of their magnitudes ($\Vert \boldsymbol{g}_u \Vert \gg \Vert \boldsymbol{g}_d \Vert$ or $\Vert \boldsymbol{g}_u \Vert \ll \Vert \boldsymbol{g}_d \Vert$).
>
> Learning from the Aligned-MTL (multi-task learning), the condition number is optimal ($\kappa(\boldsymbol{G})=1$) if and only if the gradients are orthogonal and equal in magnitude which means that the system of gradients has no dominance or conflicts:
> $$\kappa(\boldsymbol{G})=1 \iff <\boldsymbol{g}_u,\boldsymbol{g}_d>=1.$$
>
> The final gradient linear system $\hat{\boldsymbol{G}}$ satisfies the optimal condition by the condition number. Thus, the feature learning constraint $\mathcal{S}(\boldsymbol{ \theta}^{\star})$ can be defined as the following optimization to eliminate training instability:
> $$ \min _{\hat{\boldsymbol{G}}}\|\boldsymbol{G}-\hat{\boldsymbol{G}}\|_F^2 \quad \text { s.t. }\kappa(\hat{\boldsymbol{G}})=1 \iff \min _{\hat{\boldsymbol{G}}}\|\boldsymbol{G}-\hat{\boldsymbol{G}}\|_F^2 \quad \text { s.t. }\hat{\boldsymbol{G}}^{\top} \hat{\boldsymbol{G}}=\boldsymbol{I}.$$
>
> The problem can be treated as a Procrustes problem and can be solved by performing a singular value decomposition (SVD) to $\boldsymbol{G}$ ($\boldsymbol{G}=\boldsymbol{U} \boldsymbol{\Sigma} \boldsymbol{V}^\top$) and rescaling singular values corresponding to principal components so that they are equal to the smallest singular value:
> $$
> 		\hat{\boldsymbol{G}}=\sigma \boldsymbol{U} \boldsymbol{V}^{\top}=\sigma \boldsymbol{G} \boldsymbol{V} \boldsymbol{\Sigma}^{-1} \boldsymbol{V}^\top,
> $$
> where,
> $$(\boldsymbol{V}, \boldsymbol{\lambda})= eigh(\boldsymbol{G}^{\top}\boldsymbol{G}),$$
> $$\boldsymbol{\Sigma}^{-1} = diag(\sqrt{1/\lambda_{max}},\sqrt{1/\lambda_{min}}).$$
> $eigh$ finds eigenvectors $\boldsymbol{V}$ and eigenvalues $\boldsymbol{\lambda}$, with $diag$ representing a diagonal matrix. $\lambda_{max}$ and $\lambda_{min}$ are the maximum and minimum eigenvalues of $\boldsymbol{\lambda}$. The stability criterion, defined by the condition number, sets a linear system to an arbitrary position scale. To resolve this ambiguity, we select the largest scale ensuring convergence to the optimum, corresponding to the minimal singular value of the initial gradient matrix:
> $$\sigma=\sigma_{\min }(\boldsymbol{G})=\sqrt{\lambda_{min}}.$$
>
> **A5:** The GMTA process operates during the computation and updating stages of two gradients, approximately every $n$ iteration (gradient update), to balance the independence and coherence of various tasks. Tab. R.2 presents the ablation analysis of the $n$ parameter. Detail analysis is in **Experiment Results part of Global Author Rebuttal**.
>
> **A6:** We have added ablation experiments on CFDP, involving whether to use CFDP and the number of proposed boxes, as shown in Tab. R.3. Detail analysis is in **Experiment Results part of Global Author Rebuttal**.
>
> **A7:** The paper designs the first end-to-end joint training paradigm in MFD. E2E-MFD enhances interaction between intrinsic features through synchronous joint optimization, streamlining the process. Our solution goes beyond task concatenation, incorporating dedicated module design. Inspired by a phylogenetic tree analogy, we employ an ORPPT to extract features across multiple region scales. By utilizing PFMM and $L$ RFRM branches, we capture various granularities from coarse to fine. Tab. 5 validates our approach, while Fig. 6 underscores the importance of effectively integrating pixel-level and object-level details.
>
> MF primarily focuses on pixel-level relationships between image pairs, while OD integrates object semantics within diverse scene contexts. This inherent optimization barrier between the tasks necessitates a solution. We propose GMTA, a gradient alignment concept within the multi-task learning framework, to optimize task dominance and resolve conflicting gradients in end-to-end joint training. Results in Tab. 4 demonstrate that GMTA, with shared weights, yields superior performance. Comparison with other methods in Tab. 6 further validates our approach. Fig. 5 illustrates how GMTA balances shared parameters between MF and OD, effectively mitigating gradient dominance and conflict.
>
> **A8:** In Tab. R.4, three recent fusion SOTA methods (CVPR2024 SHIP, PR2024 CFNet, and PR2024 DSFusion) and three evaluation metrics (Qabf, PSNR, and SSIM) are incorporated to valid effectiveness of our method. Detailed analysis is illustrated in **Author Rebuttal Experiment Results**.

---

> > ### Author Response · Authors · 2024-08-13
> > **Follow-up on discussion**
> >
> > Dear Reviewer bix9.
> > We hope that our rebuttals have clarified your concerns. if there are any specific analyses or complementary experiment that could clear your doubts, we would be happy to try and provide them. We sincerely thank you again for your time and feedback.

---

### Author Rebuttal · Authors · 2024-08-05

# Global Author Rebuttal
We thank the reviewers for their comments. We are encouraged that the reviewers appreciate the **sound technology** (6qWk, cEzg), **well-organized writing** (bix9, 6qWk, cEzg), **certain influence** (bix9, 6qWk, cEzg), **clear motivation** (bix9, 6qWk, cEzg), and **excellent experimental performance** (bix9, G1Ro, 6qWk, cEzg). All suggestions were seriously considered and we will carefully revise the manuscript. We address each reviewer in individual comments.

## Motivation and Novelty.
The paper introduces the first end-to-end joint training paradigm in the MFD field, as acknowledged by reviewer comments: reviewer cEzg finds the idea of simultaneous learning intriguing and reasonable, reviewer 6qWk notes it’s the first attempt at simultaneous single-stage training, and reviewer G1Ro acknowledges the novel approach presented in the paper.

Current research aims to enhance image informativeness and detection performance through joint learning algorithms integrating MF and OD networks. However, existing non-end-to-end optimization methods involve cumbersome multiple steps that reduce training efficiency. Therefore, we propose E2E-MFD as the first end-to-end joint training paradigm in the MFD field. E2E-MFD facilitates interaction between intrinsic features from both domains through a streamlined, one-stage process. Our approach includes dedicated module design, such as the novel ORPPT and CFDP mechanism, to effectively balance and integrate fine-grained details with semantic information at pixel and object levels. Additionally, we found the gradient conflict problem in synchronous training for the first time and designed GMTA  to optimize task dominance and resolve conflicting gradients. Comprehensive experiments show that E2E-MFD is superior to the existing pipeline.

## Experimental Results.
According to the suggestions of **reviewer bix9**, we have added the ablation experiments and detailed information is available in the **pdf document**:

A study of fluctuations in single category detection accuracy M3FD dataset is shown **Table R.1**. We prove that it is common to observe fluctuations in detection accuracy within a single category on the M3FD dataset. Despite fluctuations, our algorithm significantly outperforms others in overall mAP.

Table R.1: A study of fluctuations in single category detection accuracy.

| Method  | People | Car  | Bus  | Motorcycle | Lamp | Truck | $ \text{mAP} _ {50} $ | $ \text{mAP} _ {50:95} $ |
| :-: | :-: | :-: | :-: | :-: | :-: | :-: | :-: | :-: |
| U2Fusion |    47.5    |   69.3   | 73.0     |      43.7      | 44.9     | 62.2      |          86.9           |            56.8            |
| U2Fusion |    47.7    |   **70.1**   | 73.2     |      43.2      | 44.6     | 63.9      |          87.1           |            57.1            |
| E2E-MFD |    **60.1**    |   69.5   |  **81.4**     |      **52.2**      |  **47.6**    |  **72.2**      |          **91.8**           |            **63.8**            |


The GMTA is performed approximately every $n$ iteration (gradient update), focusing on balancing the independence and coherence of various tasks. **Table R.2** presents the ablation analysis of the $n$ parameter of GMTA. Decreasing $n$ initially disrupts task optimization due to frequent alignment, while increasing $n$ becomes crucial when the network determines task optimization directions. However, excessively large $n$ leads to significant deviations in task paths, making alignment more challenging and negatively impacting performance.

Table R.2: Ablation studies of the iteration parameter $n$.

| $n$  | EN | MI  | VIF | $\text{mAP} _ {50}$ | $\text{mAP} _ {50:95}$ |
|:-:|:-:|:-:|:-:|:-:|:-:|
| 500  | 6.17 | 15.05 | 1.58 | 90.93 | 62.93   |
| 1000 | **6.36**  | **15.47**  | **1.65**  | **91.80**  | **63.83** |
| 1500 | 6.24 | 15.08 | 1.62 | 91.10 | 63.16    |
| 2000 | 6.13 | 14.69 | 1.45 | 90.35 | 62.75  |

We conducted ablation experiments on CFDP in **Table R.3**, investigating its inclusion and the number of proposed boxes. In the setting without CFDP, we maintained the backbone network while substituting CFDP with RPN(Region Proposal Network), standard components in two-stage object detectors. Results indicate that CFDP enhances detailed information capture and precise box guidance, thereby enhancing fusion image quality and detection performance. For optimal balance between performance and efficiency, we selected 500 proposal boxes.

Table R.3: Ablation studies of the CFDP.

| Settings | Proposal boxes | EN | MI | VIF | $\text{mAP} _ {50}$ | $\text{mAP} _ {50:95}$ | Tr.Time |
|:-:|:-:|:-:|:-:|:-:|:-:|:-:|:-:|
| w/o CFDP     | 500                | 6.07   | 14.78  | 1.58    | 90.13                 | 61.98                    | 2h52m11s    |
|    w CFDP         | 300                | 6.23   | 14.97  | 1.60    | 90.89                 | 63.29                    | 2h23m45s    |
|   w CFDP          | 500                | 6.36   | **15.47** | **1.65** | 91.80               | **63.83**               | 2h50m32s    |
|   w CFDP           | 1000               | **6.37** | 15.34  | 1.63    | **92.05**             | 63.75                    | 3h32m30s    |

In **Table R.4**, three recent fusion SOTA methods (CVPR2024 SHIP, PR2024 CFNet, and PR2024 DSFusion) and three evaluation metrics (Qabf, PSNR, and SSIM) on the three datasets (M3FD, TNO, and RoadScene) are incorporated to valid effectiveness of our method. Compared with other SOTA methods, E2E-MFD achieved superior performance across multiple metrics.

Table R.4: Quantitative results of different fusion methods.

| Method| EN ↑ | MI ↑ | VIF ↑ | Qabf ↑ | PSNR ↑ | SSIM ↑ |
|:-:|:-:|:-:|:-:|:-:|:-:|:-:|
| CFNet      | 5.64 | 13.97 | 1.54 | 0.44 | 27.91 | 1.24   |
| DSFusion   | 5.93 | 13.95 | 1.57 | 0.45 | 28.12 | 1.34   |
| SHIP       | 6.19 | 15.02 | 1.61 | 0.50 | 29.25 | 1.38   |
| E2E-MFD    | **6.36** | **15.47** | **1.65** | **0.51** | **30.01** | **1.42** |

---

### Decision · Program_Chairs · 2024-09-25

**Decision:**

Accept (oral)

**Comment:**

This paper proposes a joint learning method for multimodal fusion and object detection. The proposed network achieves SOTA performance with affordable computational cost. This paper received one slight negative score and three very positive scores. All the positive reviewers were satisfied with the author response, while the negative reviewer did not reply to the rebuttal. The AC read the response and discussions and decided to accept this paper.